# LeafArea Package: A Tool for Estimating Leaf Area in Andean Fruit Species

**Pedro Alexander Velasquez-Vasconez [1,*] and Danita Andrade Díaz [2]**

[1] Escuela de Ciencias Básicas, Tecnología e Ingeniería, Universidad Nacional Abierta y a Distancia, Pasto 520003, Colombia

[2] Vicerrectoría de Inclusión Social para el Desarrollo Regional y la Proyección Comunitaria, Escuela de Ciencias Agrícolas, Pecuarias y del Medio Ambiente, Universidad Nacional Abierta y a Distancia, Pasto 520003, Colombia; danita.diaz@unad.edu.co

\* Correspondence: pavelasquezv02@gmail.com

**Abstract:** The LeafArea package is an innovative tool for estimating leaf area in six Andean fruit species, utilizing leaf length and width along with species type for accurate predictions. This research highlights the package's integration of advanced machine learning algorithms, including GLM, GLMM, Random Forest, and XGBoost, which excels in predictive accuracy. XGBoost's superior performance is evident in its low prediction errors and high $R^2$ value, showcasing the effectiveness of machine learning in leaf area estimation. The LeafArea package, thus, offers significant contributions to the study of plant growth dynamics, providing researchers with a robust and precise tool for informed decision making in resource allocation and crop management.

**Dataset:** Velasquez-Vasconez et al. (2023). Photographs of leaves from seven plant species including *S. quitoense*, *S. betaceum*, *P. peruviana*, *R. glaucus*, *P. ligularis*, *P. vulgaris*, and *P. edulis* figshare. Figure. https://doi.org/10.6084/m9.figshare.24618183.

**Keywords:** precision agriculture; crop breeding; high-throughput phenotyping; modeling techniques

## 1. Introduction

Leaf area estimation serves as a vital parameter in various agricultural practices, including crop management, yield prediction, and the optimization of resource utilization [1]. Recognizing this significance, our study introduces the LeafArea package available on GitHub (https://github.com/velasquez-vasconez/LeafArea (accessed on 12 December 2023)), a sophisticated tool tailored for the precise calculation of leaf area in six distinct Andean fruit species: *Solanum quitoense*, *Solanum betaceum*, *Physalis peruviana*, *Rubus glaucus*, *Passiflora ligularis*, and *Passiflora edulis*.

These prominent fruit species play important roles in the economy and traditional culture of the Andean region. These fruits are not only integral to the ecological diversity of the Andean region but also pivotal in the local economy and cultural traditions [2]. Their cultivation and utilization have been deeply intertwined with the livelihoods of Andean communities for generations. Andean fruit species have gained global recognition for their nutritional value, unique flavors, and potential health benefits [2–4]. Exotic fruits continue to grow worldwide; understanding the growth and productivity of these species becomes increasingly relevant. Accurate leaf area estimation, as studied in this research, provides a crucial foundation for optimizing cultivation practices, resource allocation, and ultimately enhancing the yield and quality of these valuable fruits [5]. By delving into the intricate relationships between leaf traits and area, this study not only contributes to the

scientific understanding of plant growth but also offers practical insights that can benefit both farmers and researchers working to maximize the potential of Andean fruit species.

In contrast to traditional methods like ImageJ, which require advanced techniques and the known dimensions of images for leaf area calculations, the LeafArea package offers a more user-friendly and efficient approach. It eliminates the need for taking images or photographs and does not require any advanced technique. The LeafArea program includes a database of up to 800 observations (expandable) on which to build the most adjusted models. This enables the calculation of the approximate total leaf area of these fruit species with just the measurements of leaf length and width.

Our primary objective centers on identifying the most effective model for elucidating the intricate relationship between leaf width, length, and area specific to each plant species. The LeafArea package computes leaf area using the best GLM and GLMM, as described in this paper. Additionally, it incorporates two robust machine learning algorithms, namely Random Forest and XGBoost, demonstrating its potential to revolutionize leaf area estimation practices.

Accurate and reliable models for estimating leaf area based on easily measurable leaf traits are invaluable tools for both researchers and farmers. This innovative approach not only ensures accurate leaf area estimations but also propels the study into the forefront of modern research methodologies in plant science. The LeafArea package emerges as a transformative tool, facilitating advancements in our understanding of Andean fruit plant growth and providing a valuable tool for researchers and farmers to optimize plant breeding practices and enhance productivity in the region.

## 2. Materials and Methods

The growth patterns of leaves from various plant species were evaluated, including blackberry (*R. glaucus*), tamarillo (*S. betaceum*), sweet granadilla (*P. ligularis*), lulo (*S. quitoense*), goldenberry (*P. peruviana*), and passion fruit (*P. edulis*). The plants were planted in experimental plots that were established in six municipalities of the department of Nariño: Arboleda, Sandoná, La Florida, El Peñol, Providencia, and Ipiales. To calculate plant leaf area using the ImageJ program v1.4.3 [6], digital images of the plant leaves were captured under proper scales and lighting. A total of 800 images and leaf dimension data were utilized to construct mathematical models capable of accurately estimating leaf area. Subsequently, these images were imported into ImageJ, where the user selects the region of interest by tracing the outline of each leaf. ImageJ then calculates the area of the selected region of interest, providing an accurate measurement of the leaf area in pixels. To convert this measurement to a physical unit, such as square centimeters, a scale calibration was performed using a reference object of known dimensions within the image. Finally, the software provides the calculated leaf area in the desired unit, allowing for the precise and efficient analysis of plant leaf size. We initiated our analysis by creating a pairwise scatter plot matrix, which provided insights into the relationships between leaf area, leaf length, and leaf width. To address the observed non-linear relationships between leaf area and its predictors, we employed a square root transformation (sqrt) on the response variable. This transformation was applied to enhance the functional form of the variable and to achieve better data symmetry. The models incorporated species type, leaf length, and leaf width as independent variables. A high degree of correlation between the leaf length and width variables indicated the presence of multicollinearity issues. Variance inflation factors (vif) exceeded 15 for the leaf dimension variables, suggesting potential problems in statistical analysis. To mitigate these issues, we adopted a common practice of retaining the predictor variable that demonstrated the best model fit. Subsequently, we decided to eliminate the leaf width variable, which reduced multicollinearity in the final model. As an additional strategy, we introduced a synthetic variable, denoted as 'Length_width', which was computed as the square root of the product of the leaf length and width. In addition to deterministic models, we assessed machine learning techniques such as Random Forest and XGBoost. Thus, our methodology encompassed four ML models: a standard linear model, a Random Forest model with 500 trees and 1–10 mtry values using repeated cross-validation,

an XGBoost model with specific hyperparameters like learning rate (0.1) and max depth (6), and a linear mixed model using the REML method with random effects. For each model, comprehensive performance metrics such as the RMSE, MAE, MAPE, R-squared, and MSE were calculated. The package's functionality was demonstrated through the calculation of leaf areas, with specific configurations and hyperparameters tailored to each model. The data were then split into training and testing sets with a random seed set to ensure reproducibility. The splitting ratio was 80:20 for the training and testing sets, respectively. The Random Forest and XGBoost models were implemented using the 'rf' and 'xgbTree' methods, respectively, from the 'caret' package [7]. Comprehensive details of the coding aspects of each machine learning model utilized in this study are thoroughly documented in our online repository. For an in-depth understanding of these configurations and the adjustment procedures implemented in our analysis, readers are encouraged to visit https://github.com/velasquez-vasconez/LeafArea/tree/master/R (accessed on 12 December 2023). The performances of the four models were evaluated on the test and training sets. Finally, the best models were implemented in the LeafArea package to predict the leaf area for the entire dataset, and the predictions were added as new variables to the original dataset. All statistical procedures were performed using R software v4.2.3 [8].

## 3. Results and Discussion

The pairwise scatter plot matrix revealed that leaf area revealed a significant positive correlation ($p < 0.001$) with both leaf length and width (Figure 1). As expected, the expansion of the leaf surface demonstrates exponential growth in relation to the independent variables (Figure 1). As the leaf continues to grow, especially in terms of both width and length, the rate at which its area increases accelerates significantly. To address the observed non-linear relationships between leaf area and its predictors, we applied a square root transformation (sqrt) to the response variable. The square root transformation improved the functional form of the variable and the symmetry of the data, as evident from the distribution of points and the boxplots (Figure 1). Furthermore, the correlation coefficient with the predictor variables improved by up to four points (Figure 1). This statistical technique is effective in cases where the data exhibit a right-skewed distribution or when the relationship between variables is curvilinear, meaning that the rate of change is not constant [9]. The square root transformation is one of the power transformations used to stabilize variances and linearize relationships [10].

The high degree of correlation between the variables 'Leaf length' and 'Leaf width' indicated the presence of multicollinearity problems. Variance inflated (vif) values were found to be greater than 15 for the leaf dimension variables (Figure S1). Multicollinearity can create problems in statistical analyses, as it becomes challenging to disentangle the unique contributions of each predictor variable to the dependent variable [11–13]. To mitigate multicollinearity problems, a common practice is to retain the predictor variable that demonstrated the best model fit and the lowest RMSE values. It was decided to eliminate the leaf width variable that generated the models with the lowest fit to reduce multicollinearity in the final model. As an additional strategy, we introduced a synthetic variable, denoted as 'Length_width', which was computed as the square root of the product of the leaf length and width. The composed variable was identified as the most suitable representation of leaf expansion and played an important role in producing the most effective GLM and GLMM models (Figure 2), as suggested by Favero [14] and Freedman [15].

The GLMM models are better suited for data with hierarchical or clustered structures, where observations are not necessarily independent [16]. The highest log-likelihood value was from the GLM3 model that provided the best overall fit among the GLM and GLMM models (Figure 2). Among the GLM models, GLM3 had the highest log-likelihood value. The obtained results emphasize the significance of the synthetic variable 'Length_width' as a more predictive factor compared to the individual variables that were independently evaluated. Synthetic variables are often created by combining or transforming multiple individual variables to better represent complex underlying relationships in the data [17]. On the other hand, GLMMs are

better suited to capturing the intricate relationships often encountered in real-world datasets (Figure 3). By doing so, they enhance the predictive accuracy and model performance [18]. This collective evidence underscores the importance of adopting comprehensive modeling approaches, such as GLMMs and composite variables, when seeking a deeper understanding of complex datasets and striving for more robust predictive capabilities.

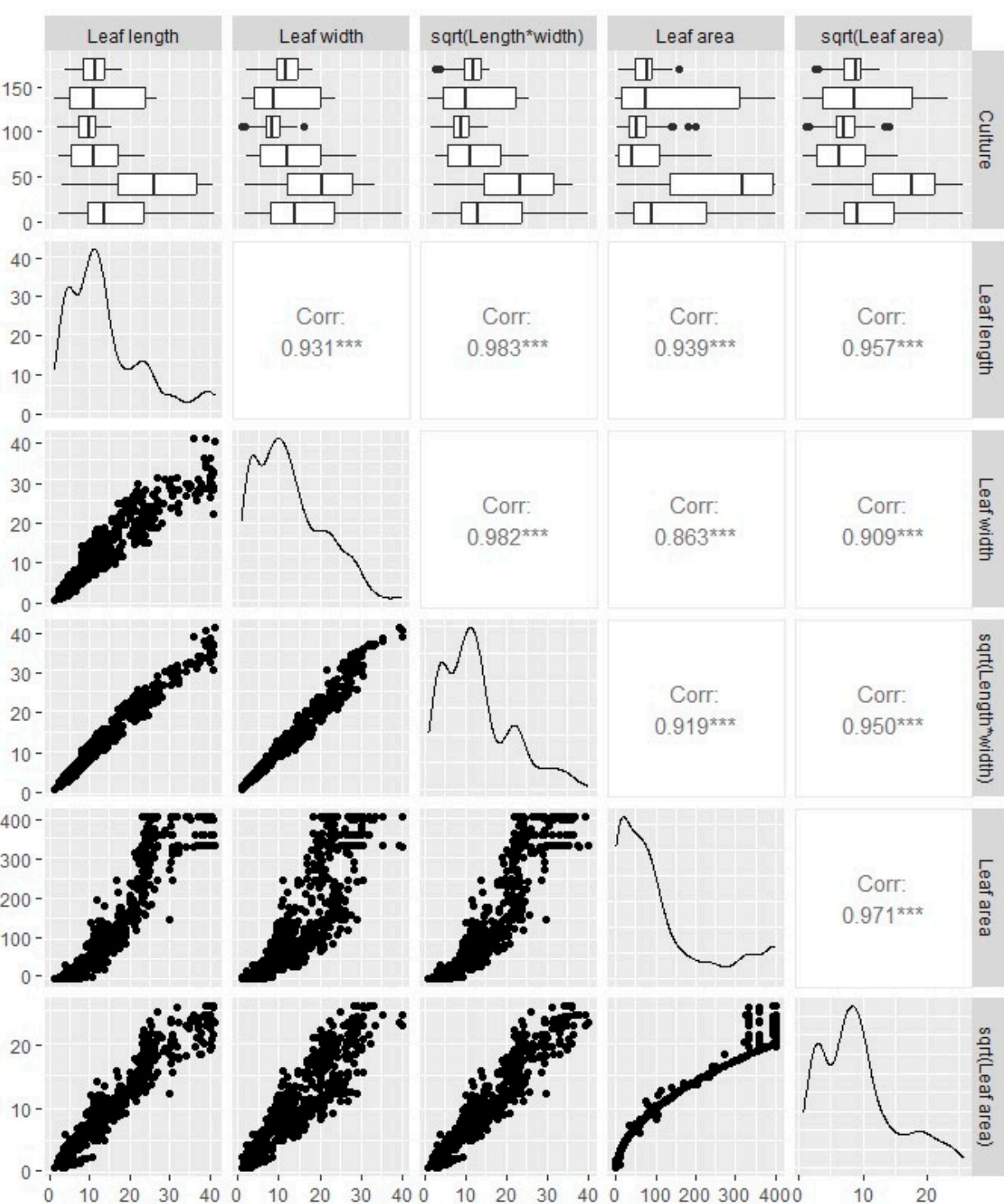

**Figure 1.** Pairwise scatter plot matrix and correlation analysis between the variables. The leaf area exhibits a significant positive correlation (*** = $p < 0.001$) with both leaf length and width. The leaf area was subjected to a square root transformation (sqrt) in response to the observed non-linear relationship. A synthetic variable was created using the square root of the product of the leaf length and width.

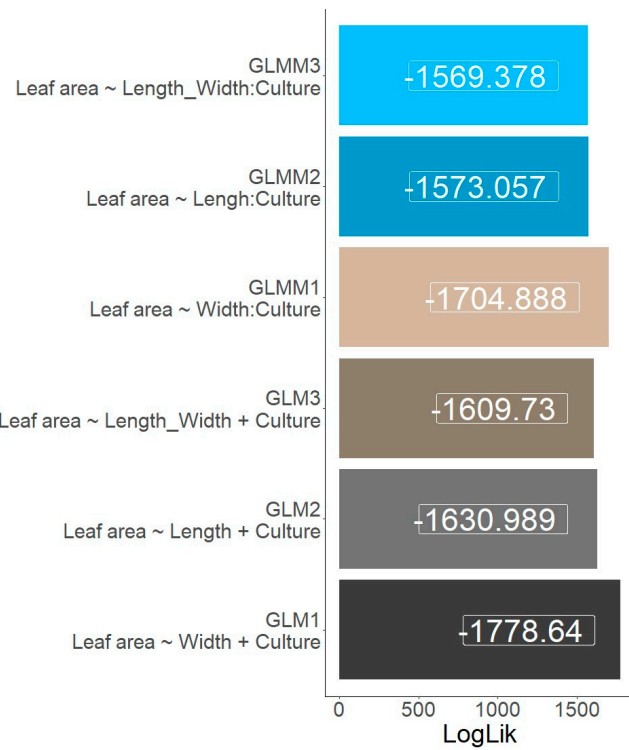

**Figure 2.** Comparing log-likelihoods between generalized linear models (GLMs) and generalized linear mixed models (GLMMs). The parameter 'Length_width' represents the composed variable obtained from SQRT (Length*Width).

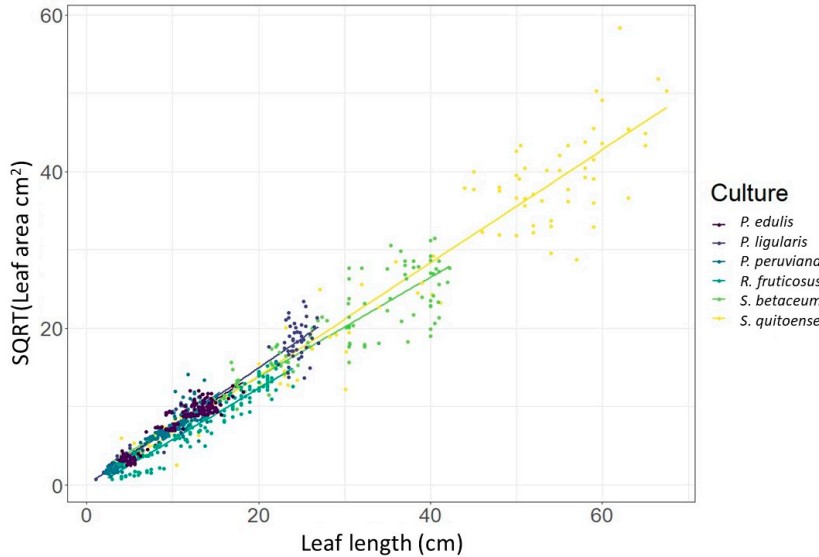

**Figure 3.** Relationship between the square root of the leaf area and leaf length in a generalized linear mixed model across six fruit species.

In addition to deterministic models, we evaluated machine learning techniques such as Random Forest and XGBoost. An evaluation of their performance metrics offered a holistic perspective on their predictive capabilities (Table 1). Notably, the results revealed a clear hierarchy of predictive power (Table 1). Among the GLMs and GLMMs, GLM3 and GLMM3 emerged as the strongest contenders, showcasing lower prediction errors and higher $R^2$ values. However, the machine learning models, particularly XGBoost, surpassed all others, exhibiting remarkably lower RMSE, MAE, and MAPE values and the highest $R^2$. This outcome underscores the remarkable potential of machine learning techniques in

enhancing predictive accuracy and highlights XGBoost as a standout performer, making it a compelling choice for tasks that demand precise and robust predictions.

**Table 1.** Performance metrics of various predictive models were compared using the test set.

| Models | RMSE | MAE | MAPE | $R^2$ |
|---|---|---|---|---|
| GLM1 | 1.8141 | 1.3980 | 22.0732 | 0.9053 |
| GLM2 | 1.5440 | 1.0946 | 18.5890 | 0.9314 |
| GLM3 | 1.4840 | 1.0651 | 16.7689 | 0.9366 |
| GLMM1 | 1.6140 | 1.1470 | 16.0398 | 0.9240 |
| GLMM2 | 1.4316 | 1.0130 | 15.7745 | 0.9390 |
| GLMM3 | 1.3946 | 0.9614 | 13.7895 | 0.9410 |
| Random Forest | 1.2099 | 0.9578 | 10.7773 | 0.9655 |
| XGBoost | 0.3043 | 0.1801 | 1.4751 | 0.9990 |

The comparison of performance metrics across various modeling techniques reveals a striking contrast, particularly with the introduction of machine learning methods like Random Forest and XGBoost into the analysis. While the traditional GLMs and GLMMs offer reasonably good predictive performances, it becomes evident that these models have certain limitations when striving for highly accurate predictions. However, with the advent of machine learning techniques, we observe a significant leap in predictive power. This remarkable outcome underscores the transformative potential of machine learning in enhancing predictive accuracy. The precision and robustness of XGBoost positioned it as a standout performer, making it an exceptionally compelling choice for tasks demanding the utmost accuracy and reliability in predictions [19]. These results not only validate the effectiveness of machine learning but also emphasize the importance of selecting an appropriate modeling approach to achieve superior predictive outcomes, particularly when working with complex or high-dimensional data.

The LeafArea package has undergone a meticulous model selection process, resulting in the identification of the optimal GLM and GLMM for calculating leaf areas across six distinct species of fruit plants. These selected models have been incorporated into a dedicated function within the package, ensuring accurate and reliable leaf area predictions (calculate_LeafArea_glm and calculate_LeafArea_glmm, respectively). Moreover, specialized functions have been developed to compute leaf area using state-of-the-art machine learning techniques, specifically the XGBoost and Random Forest models, (calculate_LeafArea_rf and calculate_LeafArea_xgb, respectively). The four functions not only provide leaf area estimates but also furnish comprehensive predictive power evaluation metrics. These metrics empower users to make informed decisions by comparing and selecting the model that best aligns with their specific requirements, thus enhancing the versatility and usability of the LeafArea package. This approach was selected due to XGBoost's exceptional capability in capturing non-linear patterns and handling multicollinearity, a common issue in biological datasets. Our findings revealed that XGBoost significantly outperformed conventional statistical models, as indicated by its lower Root Mean Square Error (RMSE) and higher coefficient of determination ($R^2$). These results highlight the efficacy of machine learning techniques, particularly XGBoost, in providing nuanced insights into plant growth phenomena, thereby offering a substantial contribution to the field of quantitative botany.

The four functions have been implemented in the R LeafArea package for calculating leaf area, currently for six plant species. We encourage researchers to provide sufficient data to expand both the number of species and the number of observations, thereby continually enhancing the predictive power of our models. This includes broadening the range of plant species that can be studied. The LeafArea package is open-source (https://github.com/velasquez-vasconez/LeafArea (accessed on 12 December 2023)), and any contributions to the database or code will be greatly appreciated.

## 4. Conclusions

Our study presents the LeafArea package as a groundbreaking tool for estimating leaf area in six Andean fruit species, leveraging a dataset of over 800 observations to construct highly accurate models. This package, a significant advancement in precision agriculture, combines traditional statistical models (GLM and GLMM) with advanced machine learning algorithms (Random Forest and XGBoost), with the latter demonstrating superior performances in terms of predictive accuracy. In particular, XGBoost's exceptional capabilities in handling non-linear patterns and multicollinearity, evident from its lower RMSE and near-perfect $R^2$ value, highlight its potential in transforming leaf area estimation practices. Our results strongly support the integration of machine learning techniques in agricultural research, offering insights that are more nuanced and robust than those provided by conventional methods. The LeafArea package actively encourages collaborative contributions to its database and code, fostering a collective effort to advance our comprehension of plant growth dynamics and productivity.

**Supplementary Materials:** The following supporting information can be downloaded at: https://www.mdpi.com/article/10.3390/ijpb15010009/s1, Figure S1: Diagnostic plots for regression model assessment using the check_model function.

**Author Contributions:** Conceptualization, P.A.V.-V. and D.A.D.; methodology, P.A.V.-V. and D.A.D.; software, P.A.V.-V.; validation, P.A.V.-V. and D.A.D.; formal analysis, P.A.V.-V.; investigation, P.A.V.-V. and D.A.D.; resources, D.A.D.; data curation, P.A.V.-V.; writing—original draft preparation, P.A.V.-V.; writing—review and editing, P.A.V.-V. and D.A.D.; visualization, P.A.V.-V.; supervision, D.A.D.; project administration, D.A.D.; funding acquisition, D.A.D. All authors have read and agreed to the published version of the manuscript.

**Funding:** The study was funded by the Sistema General de Regalias (SGR) of the Ministerio de Ciencia Tecnología e Innovación, Colombia (MINCIENCIAS). Research project: "ESTUDIO DE SISTEMAS DE CULTIVO ASOCIADOS A LOS FRUTALES ANDINOS ESTRATEGIA INNOVADORA PARA LA REACTIVACIÓN ECONÓMICA DE LOS MUNICIPIOS DE SANDONÁ, LA FLORIDA, ARBOLEDA, PROVIDENCIA Y EL PEÑOL" with BPIN number 2020000100677.

**Data Availability Statement:** The data are available in the 'dataset' section and on the provided GitHub page (https://github.com/velasquez-vasconez/LeafArea).

**Acknowledgments:** The authors would like to extend their heartfelt gratitude to Johana M. Belalcazar, Jenifer B. Vargas, Laura M. Pantoja, Luisa F. Vallejo, Javier M. Chamorro, Carlos Charfuelan, and Tania M. Pantoja for their invaluable support and dedication in collecting and organizing the data for this study; their contributions are deeply appreciated. The authors are deeply grateful to Martha I. Cabrera Otalora and Juan S. Chirivi Salomon for their invaluable support and guidance.

**Conflicts of Interest:** The authors declare no conflicts of interest.

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
