# Peer review of "LeafArea Package: A Tool for Estimating Leaf Area in Andean Fruit Species"

_2037-0164, doi:10.3390/ijpb15010009_

Round 1

Reviewer 1 Report

Comments and Suggestions for Authors

This paper introduces a toolkit called LeafArea, which is used to calculate leaf area of six types of Andean fruit trees. The toolkit utilizes machine learning algorithms to accurately estimate leaf area based on variables such as leaf length and width. The researchers tested four different machine learning algorithms and found that XGBoost algorithm had the highest prediction accuracy. This study highlights the potential of machine learning technology in accurately and reliably estimating leaf area. However, this manuscript lacks clear descriptions of some method details, and the practical application value of this method is also questionable. Only by reasonably addressing the following questions can the likelihood of publication be increased.

1. Since leaf length, width, and area can be calculated using ImageJ, the LeafArea toolkit does not provide more information compared to ImageJ. The applicable scenarios and application value of the toolkit need to be further explained.

2. How many training samples were used? Also, the description of the model features are not clear. How many features were used in total? It seems that only leaf length, width, and the transformed form of their product were used.

3. How many times was the machine learning model cross-validated? Is Table 1 in the article for cross-validation results of the training samples or prediction results of the test samples? Both evaluations need to be provided in the manuscript.

4. What kind of input data is required for the practical use of LeafArea? Is it to directly predict new data using a pre-trained model or is it necessary to retrain the model?

5. If users want to retrain the model using new data, what is the minimum number of training samples required to achieve the desired prediction performance?

6. Besides the mentioned species, is LeafArea also applicable to other species?

7. Why is the XGBoost algorithm more accurate compared to other algorithms? This needs further discussion.

Comments on the Quality of English Language

The overall quality of English language in this manuscript is satisfactory.

Author Response

Dear reviewer. Thank you very much for his contributions. The answer to each point is in the attached document.

Reviewer 2 Report

Comments and Suggestions for Authors

This paper seems to be an instruction for LeafArea package. Meanwhile The mentioned ML algorithms are widely accepted, which had been involved in several Python Packages. Generally, the contribution and novelty of this study is poor.

Author Response

Based on the suggestions received, we have meticulously revised our article to clearly articulate the dual functionality of the LeafArea package, emphasizing its role not just as a statistical tool, but also as an advanced solution for calculating leaf area values. This package stands out due to its ability to generate precise leaf area estimates by simply requiring inputs of leaf length, width, and the specified plant species. This modification in our manuscript highlights the package's ease of use and its applicability in practical agricultural scenarios, particularly for the six Andean fruit species under study. By incorporating over 800 data points into its robust framework, the LeafArea package adeptly combines Generalized Linear Models (GLM), Generalized Linear Mixed Models (GLMM), and sophisticated machine learning algorithms like Random Forest and XGBoost. This integration not only underscores the package's advanced predictive accuracy but also its novelty in the realm of agricultural science. The article now clearly delineates the package's dual nature: as a comprehensive statistical model and as a user-friendly tool for direct leaf area calculation, thereby enhancing its relevance and usability for both researchers and practitioners in the field.

Reviewer 3 Report

Comments and Suggestions for Authors

The article introduces the LeafArea package, a tool for estimating leaf area in six Andean fruit species using machine learning algorithms. The authors compare the performance of four models: generalized linear model (GLM), generalized linear mixed model (GLMM), Random Forest and XGBoost, and find that XGBoost is the most accurate and robust method.

The article is well-written, clear, and informative. The authors provide a comprehensive background on the importance of leaf area estimation for plant growth and productivity and the significance of Andean fruit species for the economy and culture of the region. The authors also explain the data collection and analysis methods in detail and justify their choice of models and variables. The article presents relevant and convincing results supported by tables, figures, and metrics. The article also discusses the limitations and implications of the study and invites collaborative contributions to the LeafArea package.

The article could be improved by addressing some minor issues, such as:

The abstract could be more concise and avoid repeating information from the main text.

The introduction could provide more context on the existing methods and challenges of leaf area estimation and the gap the LeafArea package aims to fill.

The results section could provide more interpretation and discussion of the findings rather than just reporting the numbers and metrics.

The conclusion could highlight the study's main contributions and novelty and suggest future research directions.

Final conclusions should be reworded. theses should be supported by data from research results.

It is worth expanding the cited literature with the most recent items on the research topic in question.

Comments on the Quality of English Language

The article is written in a clear and formal academic style, using appropriate terminology and citations.

The article follows the structure of a technical note, with sections such as abstract, introduction, materials and methods, results and discussion, conclusions, and references.

The article uses passive voice and third-person perspective, which are common conventions in scientific writing.

The article employs various statistical and mathematical tools, such as square root transformation, variance inflation factors, generalized linear models, generalized linear mixed models, random forest, and XGBoost. These tools are explained and justified in the text, and their results are presented using tables, figures, and metrics.

The article could improve its readability by using more headings and subheadings to organize the content, especially in the results and discussion section, which is quite long and dense.

The article could also benefit from proofreading and editing, as there are some grammatical errors, typos, and inconsistencies in the text. For example, the article sometimes uses "leaf area" and sometimes "LeafArea" without explanation, and the species names are not always italicized.

Author Response

Dear reviewer, thank you very much for all your contributions. They were extremely important in improving the document. They were all done as suggested.

Round 2

Reviewer 1 Report

Comments and Suggestions for Authors

The main comments I made have been revised and there are no further suggestions.

Author Response

Great! If you have any other questions or need assistance with something else, feel free to ask!

Reviewer 2 Report

Comments and Suggestions for Authors

The changes should be highlighted in the revised version. It is difficult to find the modification of this technique note.

Please present the configuration requirements for using the proposed LeafArea package. And the setting of hyperparameters in ML algorithms should be given.

Author Response

Thank you for your valuable suggestions regarding the LeafArea package. 

Round 3

Reviewer 2 Report

Comments and Suggestions for Authors

Accept in present form.